# Eye health knowledge, attitude, and practice among special school managers and barriers to eye health programmes in special schools in Hyderabad, India

Winston D. Prakash[1,2,3]*, Priya Morjaria[2], Ian McCormick[2], Rohit C. Khanna[1,4,5]

1 Allen Foster Community Eye Health Research Centre, Gullapalli Pratibha Rao International Centre for Advancement of Rural Eye care, L V Prasad Eye Institute, Hyderabad, Telangana, India, 2 International Centre for Eye Health, London School of Hygiene and Tropical Medicine, Keppel Street, London, United Kingdom, 3 Brien Holden Institute of Optometry and Vision Science, L V Prasad Eye Institute, Hyderabad, Telangana, India, 4 School of Optometry and Vision Science, University of New South Wales, Sydney, Australia, 5 School of Medicine and Dentistry, University of Rochester, Rochester, New York, United States of America

* winston@lvpei.org

**Data Availability Statement:** The data supporting the findings of this study are openly available in

## Abstract

### Background

Children with special education needs (SEN) are at high risk of developing vision problems. In India, there is no data available on the awareness level of eye health needs of children with SEN among special school managers (SSM) and on the barriers to providing eye care for these children in schools. This study aimed to identify the awareness level among SSM and the barriers to organizing School Eye Health (SEH) programmes in special schools, as reported by the eye health program organizers.

### Methods

A mixed-method study was conducted between July and August 2020 among SSM and eye health programme organizers from a local eye care provider in Hyderabad, India. SSM participants completed an online questionnaire assessing their knowledge, attitude, and practice concerning the eye health needs of children with SEN. Quantitative responses were described with summary statistics. Qualitative interviews with eye health programme organizers were conducted via telephone, and transcripts were thematically analysed. **Results**: In total, 13/67 (19.4%) invited SSM participated and 2/4 invited eye health programme organizers (50%) were interviewed. Among the SSM participants, 92.3% were aware of vision impaired (VI) children in their schools. Awareness of potential causes of VI ranged from 53.9%-92.3%, common eye conditions ranged from 7.7%-69.2%, and difficulties experienced by children with SEN in classroom activities ranged from 46.2%-76.9%. Only 30.8% of the special schools organized SEH programmes at least once a year. Eye health programme organizers reported barriers, such as a lack of interest from SSM, unavailability of qualified screening staff, and a lack of provision for spectacles and low-vision devices.

Dryad repository (https://datadryad.org/stash/share/MP-46peyiUV9qrUdHfass2BLoM8Ankgz-hfFtwBw5fQ).

**Funding:** This study was carried out as part of MSc thesis at London School of Hygiene and Tropical Medicine (LSHTM), Keppel Street, London, UK. During the study period the principal investigator (WP) was supported by British Council for Prevention of Blindness, LSHTM, and Commonwealth eye health consortium. Other authors (PM, IM, RCK) did not receive any financial or in-kid incentives to be a part of this study. The funders had no role in the study design, data collection and analysis, decision to publish, or preparation of the manuscript.

**Competing interests:** The authors have declared no competing interest exists.

## Conclusion

This study identified varied levels of knowledge, attitudes, and practices of SSM related to the eye health needs of children with SEN. Key barriers to conducting SEH programmes included a lack of demand, inadequate human resource availability, and limited access to government-funded resources. As the study was negatively impacted by the Covid pandemic, further research with wider representation is needed to plan comprehensive eye health programmes for children with SEN.

## Introduction

Children with special educational needs (SEN) are at a high risk of developing vision impairment (VI) and ocular comorbidities [1, 2]. In India, 28% of people with a disability are under 19 years of age, and 18% of them have a vision-related disability [3]. Vision plays an important role in early childhood development, such as hand-eye coordination, communication, spatial comprehension, and social development [4–6]. Impaired vision can lead to difficulty in performing activities of daily living (ADL), anxiety, and depression [7], and, VI combined with other disabilities can amplify these difficulties further. Therefore, early identification of VI and other eye conditions in children with SEN is crucial for better health and developmental outcomes [8].

Special schools help children with activities such as communication, interaction, cognition, and learning, catering to sensory and physical needs and helping with social and mental health [9]. School eye health (SEH) programmes can be a cost-effective approach to provide children with access to high-quality eye care services [10–12]. However, their availability in schools for children with SEN is not well documented. A study by Woodhouse et al. in 2014 [13] found that only 53% of special schools in Wales, UK, had conducted eye health programmes in their schools. Donaldson et al. (2019) [14] argued that school entry vision screenings are ineffective for children in special schools in England, mainly due to the low uptake of primary care services and limited cooperation levels. However, such programs might be the only way to provide vision screening and referral services to children in low-income settings, owing to the poor parental awareness or availability of basic services. A study by Allen et al. (2021) reported on the availability of eye care services for children with SEN and found that 31% of service providers were unable to accommodate these children in their practices [15]. In India, we found no evidence of the proportion of special schools that provide eye health programmes.

Hyderabad is the capital city of Telangana state in South India. It has a population of over 10 million people, and there are over one hundred organizations involved in teaching, training, and providing services for children with SEN. According to a local eye health organization involved in delivering community-based eye health services, most SEH programmes are focused only on mainstream schools (V Sarvepally, personal email communication, 28 January 2020 –email communication available). Among studies on the knowledge, attitude, and practice (KAP) of mainstream schoolteachers about the eye health needs of children, studies in India have shown higher levels of awareness compared to other countries [16–19]. However, there is a lack of similar studies reporting on the level of awareness about the eye health needs of children with SEN and KAP among special school managers (SSM). Studies have shown that an increase in eye health awareness levels among the school staff, children, and parents leads to an increase in service uptake [20, 21]. Therefore, this study aimed to identify the level

of eye health awareness among SSM and the barriers to organizing eye health programmes in special schools from the perspective of eye health programme organizers.

## Methods

### Ethics

Ethics approval was obtained from the London School of Hygiene & Tropical Medicine's MSc Research Ethics Committee, London, UK, and the Hyderabad Eye Research Foundation, Hyderabad, Telangana, India. Participation was voluntary, and none of the participants were given any financial or other incentives for participating. All participants read and signed an informed consent form in English or Telugu before enrolling. Participants were informed that all responses would be anonymized for reporting.

### Study design, participants, and recruitment

This was a mixed-methods study comprising quantitative and qualitative components, including an online questionnaire to assess KAP among SSM and a semi-structured telephonic interview, to understand the barriers to organizing SEH programmes in special schools. Representatives of special schools, identified through a local tertiary eye care centre database completed the online questionnaire (**Fig 1**). Contact details for all identified special schools in Hyderabad were obtained from the eye care centre database and a publicly available free online resource (www.nayi-disha.org). Schools with active contact details were invited to participate in the study with telephone calls and emails.

Participants of the qualitative component included programme organisers from eye care organizations involved in community outreach activities in urban and suburban areas of Hyderabad. Invitations to participate were distributed via email, and those who responded were included in the study.

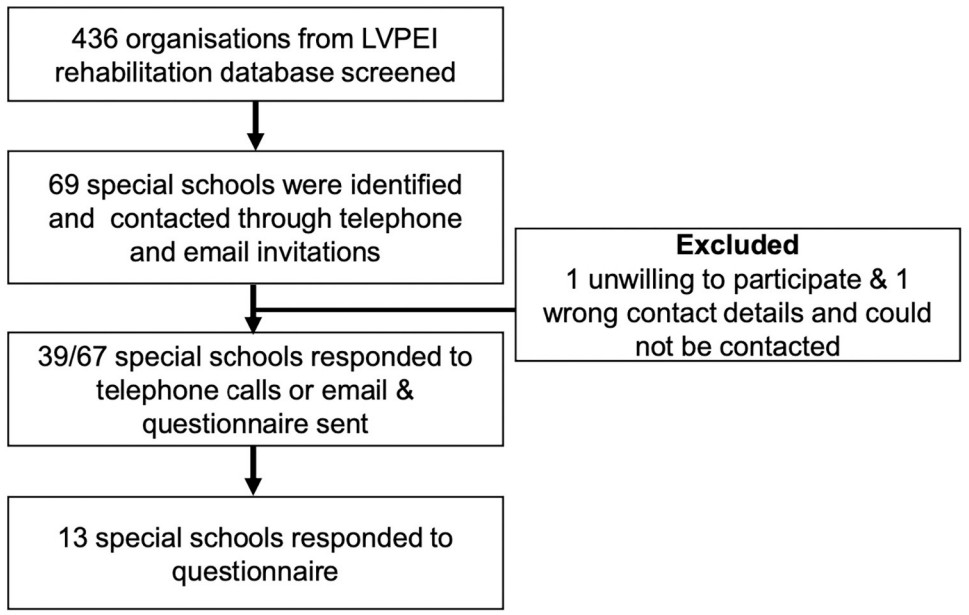

**Fig 1. Participant recruitment flow chart.**

### Inclusion and exclusion criteria

Any special school in our area of interest with active contact details was eligible for inclusion. Organizations such as early intervention centres, therapy centres, homes for the disabled and schools for the blind were excluded. Within schools, only headteachers, directors, or principals who were directly involved in school management activities (referred to in this study as 'managers') were eligible for inclusion. Any organization that was actively conducting community eye health programs in the Hyderabad region was eligible for inclusion. Within these organizations, any SEH programme organizer with a minimum of two years of experience was eligible for inclusion.

### Data collection

Data collection was carried out between July and August 2020. The participants were emailed a link to an online questionnaire prepared using Google Forms. The questionnaire included questions written for this study and some used in previous studies conducted in mainstream schools [16–19, 22]. The questionnaire included questions about the demographic details and was structured to capture KAP with regard to the eye health needs of the children attending special schools. It included multiple-choice questions and open-ended questions to allow participants to express their views (**S1 Questionnaire**). Participants were asked to complete the questionnaire within two weeks and not to refer to the Internet to ensure that their answers reflect their personal understanding. Schools that did not respond to calls or emails after three attempts were considered non-responders. Those who agreed to participate and received the questionnaire but did not complete it within two weeks were also considered as non-responders.

Programme organizers were contacted via telephone to arrange an interview. The information and consent forms were sent in advance via email. The interview guide consisted of demographic details and semi-structured questions addressing the barriers to organizing programmes in special schools. Those who did not respond to the invitation were considered non-responders.

### Data analysis

Responses to the online questionnaire were downloaded and transferred to an Excel spreadsheet (Microsoft, 2019). Descriptive summary statistics for participant demographics and MCQ responses were calculated in Excel. Any responses in open text boxes were categorized under common subthemes and listed as summary statistics in the tables. These responses are presented as themes and verbatim texts to highlight the quotes. The interviews were recorded, transcribed and coded using the inductive method. For the thematic analysis, each response was familiarized, and codes were created for emergent themes, as described by Braun and Clark [23]. These themes were further categorized under common categories of subthemes for interpreting barriers and suggestions [23].

## Results

### Questionnaire responses of special school managers

We identified 67 special schools, but only 39 could be contacted. One was public, and 38 were private schools. Although all 39 schools agreed to participate, only 13/39 (33.0%) went completed the questionnaire within the allotted time (Table 1).

**Table 1. Demographic details of school management participants.**

| Demographics | Median (n = 13) | Inter Quartile Range |
|---|---|---|
| Age | 40 | 16.5 (35, 51.5) |
| Sex | Total number of participants n = 13 (%) | |
| Male | 4 (31) | |
| Female | 9 (69) | |
| Speciality of schools | Total number of schools n = 13 (%) | |
| Multiple disability | 10 (77) | |
| Autism | 2 (15) | |
| Intellectual disability | 1 (8) | |
| | Median (n = 13) | Inter Quartile Range |
| Number of children in schools | 36 | 26 (14,40) |

## Knowledge

Most respondents [92.3% (n = 12)] were aware of the number of children with eye conditions in their respective schools, and 76.9% (n = 10) were aware that early treatment could result in better visual outcomes. However, only 23.1% (n = 3) had any knowledge that children with SEN have an increased risk of developing eye conditions. Sixty-three percent (n = 9) said they identified vision problems by just observing the behaviour of the child in classrooms.

The main emergent themes in the open-ended text box responses with regards to symptoms of reduced vision faced by the children in the classroom were *"Difficulty in reading, writing, and difficulty in copying from the blackboard,"* which was reported by 53.8% (n = 7) of participants, followed by *"Poor hand-eye coordination,"* reported by 30.8% (n = 4). Another major theme that emerged was *"Difficulty in sports activities,"* which was reported by 30.8% (n = 4). Most participants reported lack of nutrition (92.3%, n = 12) and hereditary predisposition (76.9%, n = 10) as the main reasons for children developing eye problems, and 30.8% (n = 4) felt that an eye check-up should be considered only if necessary. The most well-known condition was squint (69.2%, n = 9), and the least known was glaucoma (7.7%, n = 1) (Table 2).

## Attitude

Most participants (92.3%; n = 12) felt that children with vision problems were less capable of performing academic activities, although 69.2% (n = 9) felt that they would be able to participate in extracurricular activities. In response to the open-ended question on possible reasons why children should/should not be attending mainstream schools, a major theme that emerged was *"Inclusive schools."* Close to 2/3rd of the participants (61.54%, n = 8/13) mentioned that children with VI can interact normally with their peers and comprehend like those without VI. Additional responses were that they will also feel motivated by being with their normal peers, hence, attending mainstream schools would benefit them. On the contrary, 1/3rd of the participants felt that those with VI will benefit from special schools as they will receive additional support and care in these schools. Another third of the participants (30.8%; n = 4) felt that children with VI should attend regular schools if they are inclusive, while 15% (n = 2) said they should attend normal schools if the schools provided braille training.

A participant stated, *"The teachers need to pay more attention while teaching these children, which is not practically possible in regular schools,"* while another participant said, *"They should go to an inclusive school, they deserve to be with normal children"* (Table 3).

**Table 2. Participant response to knowledge related questions.**

| Knowledge | n = 13 (%) |
|---|---|
| **Are you aware of the number of children with eye conditions in your school? (MCQ type)** | |
| Yes | 12 (92.3%) |
| Not sure | 1(7.7%) |
| **How likely are children with SEN to develop eye problems? (MCQ type)** | |
| Very likely | 3 (23.1%) |
| Same as children without disability | 3 (23.1%) |
| Less likely | 2 (15.4%) |
| Not sure | 5 (38.5%) |
| **Among the following eye conditions, which are the most common conditions that could affect children with SEN? (MCQ type)** | |
| Squint | 9 (69.2%) |
| Big eyeballs | 4 (30.7%) |
| Refractive error | 3 (23.1%) |
| Cataract | 3 (23.1%) |
| Small eyeballs | 3 (23.1%) |
| Eye cancer | 2 (15.4% |
| Conjunctivitis | 2 (15.4%) |
| Glaucoma | 1 (7.7%) |
| **What do you think is/are the reason/s for children with SEN developing eye problems? (MCQ type)** | |
| Heredity | 10 (76.9) |
| Lack of Nutrition | 12 (92.3) |
| Physical illness | 9 (69.2) |
| Injury | 9 (69.2) |
| Consanguinity | 7 (53.9) |
| Curse from God | 2 (15.4) |
| **What are the symptoms that could indicate a child is having vision problem? (Open ended)** | |
| Difficulty in reading and writing | 7 (53.8) |
| Difficulty in sports activities | 4 (30.8) |
| Difficulty in mobility | 1 (7.7) |
| Difficulty in eye hand coordination | 4 (30.8) |
| Difficulty in copying from blackboard | 7 (53.8) |
| Difficulty in identifying objects around | 3 (23.1) |
| Others (avoid using computer, headache, neck pain, and low attention span) | 4 (30.8) |
| Not sure | 3 (23.1) |
| **How often is an eye check-up warranted for children with SEN? (MCQ type)** | |
| Twice a year | 5 (38.5%) |
| Once a year | 4 (30.8%) |
| Only if needed | 4 (30.8%) |

## Practice

Close to half of the participants (46.2%; n = 6) said that they would refer children who face difficulty in performing academic tasks, such as reading and writing, to an eye doctor, but with the same number were unsure of what to do or, gave generic responses, for example, "*We never faced such issues in our school*" or "*I would counsel the parents.*" However, 53.8% (n = 7) of participants said they make children sit closer to the blackboard or the instructor if they had difficulty in copying from the blackboard. One participant said,"*I would encourage the child to use his/her spectacles*," while another said, "*We allow the child to use prescribed low vision device*

**Table 3. Participants responses to attitude related questions.**

| Attitude | n = 13 (%) |
|---|---|
| **Do you think children with SEN can perform academic tasks? (Open ended)** | |
| Yes, they can perform academic tasks | 1 (7.7%) |
| No, they cannot perform academic tasks | 12 (92.3%) |
| **Do you think children with SEN are capable of participating in non-academic activities, such as sports or extracurricular activities? (Open ended)** | |
| Yes, they can participate and perform | 9 (69.2%) |
| Yes, but not fully capable | 1 (7.7%) |
| Not sure | 3 (23.1%) |
| **Can children with vision impairment attend mainstream school? (Open ended)** | |
| Yes | 8 (61.5%) |
| No | 4 (30.8%) |
| Not sure | 1 (7.7%) |
| **Do you think children with vision impairment can interact with normal peers? (Open ended)** | |
| Can interact normally | 12 (92.3%) |
| Did not answer | 1 (7.7%) |

*when needed."* Forty-six percent (n = 6) of participants said they had never organized an eye health programme in their schools, while one participant said, *"We approached a few doctors for eye camps, but they asked us to bring the children to their hospital."*

When asked about the availability of accessible facilities in their schools, 53.8% (n = 7) of participants stated that they have additional trained staff, while one participant stated, *"We do not have any child with low vision in our school."* Thirty-nine percent (n = 5) of the schools provided a nutritious diet in their schools. Twenty-three percent (n = 3) of participants stated that they provided a mixed nutritious diet, including pulses, millet, finger millet, fruits, green leafy vegetables as salads, chicken meat, boiled eggs, and milk, in their schools, while a few stated that they did not provide any food in their schools. However, two participants stated that they checked the lunch box and gave feedback/suggestions to the parents to include a nutritious diet (Table 4).

## Interview responses of eye health programme organizers

Of the four known community outreach organizers from four different organizations in Hyderabad, two from non-profit organizations responded and agreed to be interviewed. One was a public health specialist with over two decades of experience, and the other was an optometrist and head of community ophthalmology services with a decade of experience in the field. The interview transcripts were coded for thematic analysis. The three major themes were barriers to organizing SEH programs, existing good practices, and suggestions to improve services. Under each theme, sub-themes were coded and categorized.

## Barriers to organizing school eye health programmes

Responses to barriers were categorized into seven sub-themes as presented below:

- **Low priority**

Most of the special schools, covered by the respective organizers, were run by the government and were mostly located in urban areas. One organizer covered 1000 schools every year, of which only 0.5% were special schools. Another organizer covered 50 schools in a year, of which less than 2% were special schools. Both organizers provided three common reasons for

**Table 4. Participants responses to practice related questions.**

| Practice | N(%) |
|---|---|
| **What would you do if you see a child keeping books close to his/her face while reading and writing? (Open ended)** | |
| Refer to eye doctor | 6 (46.2%) |
| Modify working distance | 3 (23.1%) |
| Help in copying and writing | 1 (7.7%) |
| Did not answer | 1 (7.7%) |
| Others (Child may have near vision problem, we will test his/her vision) | 2 (15.4%) |
| **What classroom adaptations are made for vision impaired children in your school? (Open ended)** | |
| Encourage them to use prescribed optical devices | 2 (15.4) |
| Ask the children to sit closer to the blackboard or the teacher | 7 (53.9) |
| Provide large font and bright coloured materials | 3 (23.1) |
| Braille training | 2 (15.4) |
| Provide extra time to complete the task | 2 (15.4) |
| Did not answer | 2 (15.4) |
| **What would you do If you suspect a child has an eye problem? (Open ended)** | |
| Refer to eye doctor | 7 (53.9%) |
| Provide special care | 1 (7.7%) |
| Help them wear glasses | 1 (7.7%) |
| Ask them to stay home until they get better | 2 (15.4%) |
| Did not answer | 2 (15.4%) |
| **What would you do If a child has an untreatable eye condition? (Open ended)** | |
| Create adaptive environments | 4 (30.8%) |
| Refer them to doctor/rehabilitation | 4 (30.8%) |
| Not sure | 3 (23.1%) |
| Generic responses (Counsel parents, Train the child to live independently, we have not faced such problems yet) | 3 (23.1%) |
| **How often are eye health programs organised in your schools? (MCQ)** | |
| Once a year | 3 (23.1%) |
| Twice a year | 1 (7.7%) |
| Rarely | 2 (15.4%) |
| Never | 6 (46.2%) |
| Did not answer | 1 (7.7%) |
| **What are the facilities available for children in your schools with regards to accessibility? (Open ended)** | |
| Extra trained caretaker staff | 7 (53.8%) |
| Railings, ramps, and support handles | 2 (15.4%) |
| Comfortable seating | 1 (7.7%) |
| None | 2 (15.4%) |
| Did not answer | 1 (7.7%) |
| **Do you provide nutritious diet for children in your school? (Open ended)** | |
| No such provisions at the school | 6 (46.2%) |
| Suggest parents to include nutritious diet | 2 (15.4%) |
| Provide nutritious diet at school | 5 (38.5%) |
| None | 2 (15.4%) |
| Did not answer | 1 (7.7%) |

the low priority given for including special schools in their programmes. Firstly, both said that the reason for the exclusion of special schools in their respective programmes was that they assumed, but were not sure, that an eye examination was carried out as part of an annual general health check-up conducted in special schools. Secondly, the time taken to complete a SEH programme in one special school took longer than in a mainstream school. Finally, there is a limited time frame for completing these programmes within a school calendar year due to frequent public holidays, school vacations, and exam periods.

- **Lack of initiative**

Special SEH programmes are usually conducted as stand-alone programmes and rarely included in government or NGO-sponsored activities. The focus of their activities is on the number of children screened as opposed to needs, resulting in these organizations performing very limited targeted screenings. Respondents further claimed that, special schools seldom take the initiative to organize eye health programmes, with there being a general lack of motivation from all the stakeholders concerned. One interviewee said, "*Very rarely, we receive requests from one or two private schools to conduct eye health programmes in their schools, but never from special schools.*"

- **Stakeholder-related challenges**

Special schoolteachers are occupied with various responsibilities related to governmental/ non-governmental activities and are therefore unwilling or lack motivation to take up additional responsibilities. Without the involvement of schoolteachers, it is impossible to examine children with SEN, which is another reason reported by the organizers for not considering special schools in their programmes. They further highlighted that in some situations, the teachers or the staff are often transferred to a different school or are unavailable, which makes it difficult to convince the new management or teachers to help in organizing such programmes in these schools. Unlike mainstream schools, special SEH programmes require the parents of all the children to attend the eye examination a task extremely difficult to coordinate. Therefore, there is limited opportunity to educate and create awareness among parents about their child's eye condition and needs.

- **Skilled human resource shortages**

The organizers mentioned that special schools require a skilled workforce trained for the assessment of children with SEN, low vision assessment, and prescription of spectacles and low vision devices (LVDs). Additionally, they stated that most SEH programmes are conducted by community eye health workers, schoolteachers, and vision technicians who are not trained to perform a complete eye examination. They reported that having a more skilled workforce, including optometrists or ophthalmologists, would incur additional costs and, that in some circumstances, such cadres are unwilling to participate in the SEH programmes due to long-distance travel and limited financial benefits. Therefore, most special SEH programmes are conducted by unsupervised optometry trainees.

- **Impossible to task-shift to unskilled human resource**

Training schoolteachers, community eye health workers (CEHW), and administrative staff in screening children with SEN is not possible, as it requires more time and expertise in the field. One interviewee said, "*I can understand the difficulties in assessing these children, owing to the training that I received during my low-vision assessment training, which was part of my undergraduate degree, but not everyone can understand. So, I think having a basic training in assessing these children helps in organizing such programmes.*" Children with SEN require

additional tests for vision assessment and refraction, which cannot be performed by school staff and CEHW. In addition, there is a limited opportunity for training children in the usage of spectacles or LVDs due to the unavailability of parents, time, and training environment.

- **Additional clinical equipment and assessment needs**

The organizers mentioned that a complete eye examination for children with SEN requires expensive equipment, increasing the demand for financial resources. They also stated that only dry refraction can be performed on the school premises, and children require a referral for cycloplegic refraction. Further, poor referral uptakes among this group makes it difficult to provide effective refractive error correction via the SEH programmes alone.

- **Poor availability and affordability of spectacles and low-vision devices**

Interviewees said that the low-cost spectacles that are commonly provided through SEH programmes are not of good quality. Therefore, there is a higher chance of breakage and discomfort to children. However, providing good-quality spectacles requires increased financial resources which combined with the unavailability and unaffordability of LVDs pose a major challenge. Most of the time, these devices are just prescribed and not provided. Therefore, it is highly likely that the parents do not purchase them. One organizer said, *"Although governmental provisions are available for purchasing LVDs, these are available only on paper and not accessible in reality."*

- **Good practices and suggestions**

The two sub-themes of existing good practices observed in the included special schools and suggestions from the SEH programme organizers, for improved service provision are summarized in Table 5.

One organizer mentioned, *"Many of these special schools maintain health records for each child, which is helpful for us to understand the overall health of the child. It would be good if every school followed this habit."* According to both organizers, the screening team apply several good practices, With one stating, *"We make sure that the parents understand the condition of their child's eye, and it is also easy for us to create awareness if the parents are present during examination."*

An organizer said, *"Before every school screening program in special schools, the parents must be informed and encouraged to be present during the program. This will also improve the child's*

**Table 5. List of existing good practices and points to be considered in the future programmes.**

| Good practices followed in special schools |
| --- |
| • Medical history record keeping for every child–helps in understanding the overall medical conditions of children |
| **Good practices followed by eye care providers** |
| • Helping the parents understand the eye health needs of their children during eye examination |
| • Creating awareness among the schoolteachers about the need for eye care services for children with SEN |
| • Mandatory training of screening team members prior to the programme in examining children with SEN |
| **Points to be considered before organizing SEH programmes in special schools suggested by the SEH programme organizers** |
| • Ensure that the programme venues are accessible |
| • Inform parents/guardians well in advance and encourage them to be available during the screening |
| • Develop a standard operating protocol pertaining to special school eye health programmes |
| • Stakeholder involvement should include local NGOs and volunteers |
| • Facilitate the availability of good quality spectacles and LVDs |
| • Government should consider monetary or non-monetary incentives for participating NGOs/organizations |

*cooperation level."* Regarding financial incentives, an organizer stated, *"If the government could give some financial or other incentives to the volunteering NGOs, they will be motivated to participate in special school screening programs."*

Another organizer said, *"We ourselves find it difficult to procure low vision devices if needed as these are not easily available and are also very expensive."*

## Discussion

Universal eye health and Sustainable Development Goal 3, "to ensure healthy lives for all people at all ages," cannot be achieved without ensuring equitable access to eye health care for underserved groups [24]. This is the first study in India to report on the knowledge, attitudes, and practices (KAP) of special school managers regarding the eye health needs of children with SEN. Overall, the findings point to varied levels of understanding SSM; however, heads of special schools can be the catalysts for change in their schools and among their peers. Existing evidence also suggests that increased eye health knowledge leads to better health-seeking behaviour among parents and school management.(18, 19)

Most participants said they were aware of the number of children in their schools with eye conditions and that early intervention could result in better eye health outcomes. However, this claim could only be validated by identifying the true numbers and conducting a prevalence survey or vision assessments in these schools. In addition, if this is the case, then educating the staff to encourage/remind the parents to take their children for regular eye check-ups would enable the continuum of care for children with eye conditions. This is especially important if there are no eye health programs organized in schools. Even though most participants knew that children with SEN needed regular eye check-ups, less than a third of the schools reported conducting eye health programmes, which is lower than the 53% reported in a similar study from Wales, UK, in 2014 [13].

Although refractive error is one of the most common conditions reported in children with SEN [25, 26], only 23% of participants were aware of this condition. This is alarming, as uncorrected refractive errors could lead to avoidable VI, which would further amplify any difficulties caused by a pre-existing disability. However, if corrected in time, it could improve the quality of life for these children [27–29]. Most participants said poor nutrition could cause eye problems, but none of the participants mentioned Vitamin A deficiency (VAD). Noting the importance of vision and ocular health, perhaps including VAD specifically in the questionnaire MCQs would have impacted the responses on the causes reported. Few studies have reported VAD among children with SEN [25, 30]. This also warrants more careful selection of questions in future studies and active inclusion of health promotion as part of special SEH programs.

Most participants exhibited a negative attitude towards the ability of vision-impaired children to perform academic tasks. This could negatively impact eye health-seeking behaviour or referral practices among school staff. In many circumstances, management of a child's overall disability status takes precedence over eye health-related issues, which affects eye health-seeking behaviour among the stakeholders [31]. Almost two-thirds of the participants thought that children with only a vision impairment could study in mainstream schools, assuming that these schools are inclusive. The positive impact of inclusive education on psychosocial well-being is supported by studies from high-income countries [32]. However, the evidence in low- and middle-income countries (LMICs) is inadequate [33], and this study did not explore any further to understand this factor.

Practices such as providing non-optical assistive devices, braille printed reading materials, and encouraging children to use their spectacles and prescribed LVDs are encouraging and should be included in health education programs. The varied level (mostly low) of knowledge

about the eye health needs of children with SEN among SSM in Hyderabad is comparable to reports from mainstream primary schoolteachers outside of India [17, 18, 34]. However, it does not correlate with mainstream primary schools in India [16, 19], which could be due to the higher coverage of mainstream schools by SEH programmes compared to special schools. This might have had a positive impact on the awareness level among mainstream schoolteachers, which was also reported in China and Vietnam [35, 36]. However, this disparity could also be due to the inclusion of school heads in this study rather than school/class teachers.

It was reported that regular health check-ups in the schools included eye health, but this could not be verified in this study. Therefore, future research should prioritize assessing the frequency and ophthalmic content of general health check-ups at special schools and the competency levels of those who perform these examinations. Should the check-ups be insufficient to fully meet the needs of children with SEN, both schools and programme organizers will have a greater incentive to prioritize the inclusion of special schools in broader programme activities. It is also possible that overall disability takes priority over eye health, resulting in schools not taking the initiative in organizing SEH programmes. This also shows a possible knowledge gap among the school management.

The barriers reported for implementing programmes in special schools included a lack of initiative, scarcity of skilled human resources, difficulty in task-shifting, unaffordability, and limited availability of LVDs. A systematic review from 2018 reporting on the interventions to improve school-based eye care in LMICs also reported similar findings [37]. Training of skilled eye health workers, such as optometrists, in assessing children with SEN, prescribing glasses and LVDs to children, and incentivizing them to attend special school eye health programmes may help in addressing the human resource shortages [38]. In addition, facilitating the availability of LVDs and making them more accessible and affordable for these children are some of the aspects that need to be worked out at the policy level. Moreover, increasing eye health awareness among special school managers, schoolteachers, and parents would help in better uptake of services after referrals and improve the continuum of care [31, 38]. Further, there is a need for appropriate referral facilities within their proximity, for the referred children to avail these services.

Lack of stakeholder (teachers/parents/guardians) engagement in the SEH programme was another major concern reported. Although it was not tested in this study, a lack of motivation among teachers was reported on the assumption that they are already overburdened with other health programmes and responsibilities, a scenario also reported in Pakistan [39]. Zhoa et al., (2022) suggest that financial incentives to the teachers or staff for taking up such responsibilities might improve this situation [38]. However, this may not be a sustainable option, especially in LMICs. The organizers reported that there is limited opportunity to conduct eye health awareness sessions for primary stakeholders (parents/guardians), as most of them do not attend the screening programmes. This may also have a significant impact on the continuum of care due to poor referral uptake [31]. Perhaps counselling the parents via telephonic communication when a child has an eye problem or conducting eye health promotion sessions during parent-teacher meetings could help in addressing this issue.

The organizers also made several recommendations to improve the KAP of stakeholders and to increase the uptake of eye health programmes in special schools. Their suggestion to incorporate health promotion strategies within SEH programmes could make them more impactful and might indirectly address the issue of sustainability [31]. When the stakeholders are better informed about the eye health needs of these children, it would positively impact their eye health seeking behaviour, as reported in studies in mainstream schools in China and Vietnam [35, 36]. In addition, financial subsidiaries or free-of-charge spectacles/LVDs could

also reduce the burden on families who might otherwise be facing increased healthcare costs associated with caring for children with SEN [40].

## Limitations

This study should be interpreted in the context of its limitations. First, the participants recruited were the school heads from a minority of special schools in the Hyderabad area, and the response rate was very low. Therefore, the participants of these institutions may not be representative of all headteachers/schools in the area. Second, the study period was interrupted by COVID-19 travel restrictions, meaning face-to-face sensitization and enrolment of headteachers was not possible. Third, although the questionnaire was piloted among the peers of the principal investigator, it was not validated for the target audience. This might have resulted in varying levels of understanding between the participants and the exclusion of other pertinent questions. Moreover, it is difficult to ascertain whether the participants used any external resources to answer the questionnaire, which might have impacted the results.

The day-to-day engagement of the headteachers with the children in their schools was unverifiable and it might have been valuable to have recruited class teachers for more insight into the daily activities and challenges faced by the children. However, this was hampered by the pandemic, which posed difficulty in contacting the class teachers. Finally, the interviews were conducted telephonically and further exploration of the responses to the semi-structured questions was not performed. Such exploration would have highlighted several important components that were not captured in this study.

## Conclusion

This study highlights a potential gap in the KAP of special school managers in India. Measuring the eye health awareness level and KAP among schoolteachers in a larger sample would give better insights into the matter. Moreover, the inclusion of eye health promotion strategies to create eye health awareness among all the stakeholder groups would be beneficial. This study also highlights barriers to organizing SEH programmes in special schools that include accessibility and affordability of eye health services. Therefore, there is a need to conduct a comprehensive situation analysis or vision screening programme to estimate the prevalence of VI and other eye conditions, to fully understand the current need and service gap for children with SEN.

## Supporting information

**S1 Questionnaire. Participant questionnaire for the assessment of knowledge, attitude, and practice among special school managers.**
(PDF)

## Acknowledgments

The authors would like to acknowledge the help received from Ms.Mahalaxmi Mojjada for translating the consent form and questionnaire into the Telugu language and Mr. Abhinav Sekar for the final manuscript language editing. The authors also would like to thank all the participants for their valuable time responding to the questionnaire and the interview.

## Author Contributions

**Conceptualization:** Winston D. Prakash, Priya Morjaria, Ian McCormick.

**Data curation:** Winston D. Prakash.

**Formal analysis:** Winston D. Prakash.

**Investigation:** Winston D. Prakash.

**Methodology:** Winston D. Prakash, Priya Morjaria, Ian McCormick.

**Project administration:** Winston D. Prakash.

**Resources:** Winston D. Prakash.

**Software:** Winston D. Prakash.

**Supervision:** Priya Morjaria, Ian McCormick.

**Writing – original draft:** Winston D. Prakash.

**Writing – review & editing:** Priya Morjaria, Ian McCormick, Rohit C. Khanna.

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
