## [Decision Letter · Decision Letter 0]

28 Jul 2023

PGPH-D-23-01037

Eye health awareness among special school managements and barriers to organize eye health programme in special schools in Hyderabad, India

Dear Dr. Devaraj,

Thank you for submitting your manuscript to PLOS Global Public Health. After careful consideration, we feel that it has merit but does not fully meet PLOS Global Public Health’s publication criteria as it currently stands. Therefore, we invite you to submit a revised version of the manuscript that addresses the points raised during the review process.

Please see the comments from three reviewers below. All reviewers seem positive about the contribution of the study, and have provided detailed guidance on aspects that could be strengthened to increase clarity and impact of the study. Particularly, please pay attention to all concerns about any parts of the Methods section being unclear, in order to ensure that the study is fully reproducible by readers in the revised version.

We look forward to receiving your revised manuscript.

Kind regards,

Hanna Landenmark

Staff Editor

Journal Requirements:

1. Please provide additional details regarding participant consent. In the ethics statement in the Methods and online submission information, please ensure that you have specified (1) whether consent was informed and (2) what type you obtained (for instance, written or verbal, and if verbal, how it was documented and witnessed). If your study included minors, state whether you obtained consent from parents or guardians. If the need for consent was waived by the ethics committee, please include this information.

2. Please send a completed 'Competing Interests' statement, including any COIs declared by your co-authors. If you have no competing interests to declare, please state "The authors have declared that no competing interests exist". Otherwise please declare all competing interests beginning with twhe statement "I have read the journal's policy and the authors of this manuscript have the following competing interests:"

3. Please upload a copy of Figure 1 which you refer to in your text on page 5. Or, if the figure is no longer to be included as part of the submission please remove all reference to it within the text.

4. In the online submission form, you indicated that "The anonymized data can be made accessible if necessary in MS format". All PLOS journals now require all data underlying the findings described in their manuscript to be freely available to other researchers, either 1. In a public repository, 2. Within the manuscript itself, or 3. Uploaded as supplementary information.

Additional Editor Comments (if provided):

Reviewers' comments:

Reviewer's Responses to Questions

**Comments to the Author**

1. Does this manuscript meet PLOS Global Public Health’s publication criteria? Is the manuscript technically sound, and do the data support the conclusions? The manuscript must describe methodologically and ethically rigorous research with conclusions that are appropriately drawn based on the data presented.

Reviewer #1: Partly

Reviewer #2: Partly

Reviewer #3: Yes

2. Has the statistical analysis been performed appropriately and rigorously?

Reviewer #1: No

Reviewer #2: N/A

Reviewer #3: N/A

3. Have the authors made all data underlying the findings in their manuscript fully available (please refer to the Data Availability Statement at the start of the manuscript PDF file)?

Reviewer #1: No

Reviewer #2: Yes

Reviewer #3: Yes

4. Is the manuscript presented in an intelligible fashion and written in standard English?

Reviewer #1: Yes

Reviewer #2: Yes

Reviewer #3: Yes

5. Review Comments to the Author

Reviewer #1: Title: must better reflect the study which has a KAP component and barriers and enablers to eye care.

Methodological concerns: The study is described as mixed methods but the data is not managed or presented clearly according to requirements of each method. There needs to be clarification between the quantitative analysis and thematic analysis used in the qualitative part. The selection criteria applied must be more clearly articulated to clarify steps and sampling strategy eg. whether all managers or only one leader in each school were invited to participate etc. The questionnaire design must be detailed. ie. the basis for inclusion of specific questions in each area of the questionnaire and how were these validated for inclusion.

The Results for the quantitative analysis should include some inferential analysis eg. was there associations between knowledge and practices, or attitudes and practices and were these significant. Qualitative analysis results must be presented in more detail with the emergent Themes and the specific codes per theme explained and presented as such. Quantitative data is referred to as themes, hence causing confusion.

The Discussion will be strengthened if the methods and results are sorted out.

The Conclusion should highlight findings in relation to the key study objectives and recommendations.

There are minor language edits needed as tracked in the manuscript.

Reviewer #2: Main comments:

Many thanks for a thought-provoking article. Establishing a connection between the responses from the two groups of participants, namely teachers and School Eye Health (SEH) organizers, would provide valuable insights. Exploring teachers' perspectives on the SEH organizers' list of barriers and enablers could shed light on their perceptions and potential collaboration challenges. Notably, the organizers suggest that all children should undergo either annual or admission eye checks. It is essential to examine whether this recommendation is implemented in practice. If the organizers assert that SEH programs are unnecessary because eye care already takes place, it raises the question of whether this belief constitutes the primary barrier to effective eye care, particularly if annual checks or checks on admission or not occurring for everyone. In light of these considerations, the research could benefit from focusing less on theoretical assessments of the perceived importance of eye care and instead prioritize investigating the current state of eye care in India. Questions that address the percentage of children who have received comprehensive eye examinations, the timing of their most recent tests, the prevalence of spectacle usage or eye conditions among the children, and a comparison with existing studies would enhance the understanding of the current situation.

Introduction:

• School eye health (SEH) programs have the potential to provide children with access to good quality services and can be cost-effective (9-11). It would be helpful to specify which services are included in SEH programs.

• In a study by Woodhouse et al. in 2014 (12), it was found that only 53% of special schools in Wales, UK, had conducted eye health programs. It would be beneficial to include data from India on this topic, or mention if no data is available.

• In the introduction, it would be valuable to compare the prevalence of eye problems in children with autism, children with multiple disabilities, and children with no disabilities. Additional information and references from Donaldson papers and the Seeability website could be included.

• The reader needs to know what currently happens in India and in each school included in the research. It would be relevant to mention whether each child has received an eye examination or screening test, and what those tests consisted of. Details about vision tests, cover tests, ocular motility, BV tests, cycloplegia, refraction, and fundus and media examinations would be informative. Additionally, it would be helpful to clarify the procedures followed if a child is unable to respond to a letter test, such as using pictures or preferential looking tests, or referring the child to a hospital.

• Special school managements- Is this Special school managers/head teachers?

• Please check SEH written in full in the first instance.

• Sometimes SHE used.

• Throughout the paper, it should be explicitly addressed that the study assesses the knowledge and attitudes of school managers and the barriers and enablers of organizers to SEH. This should be reflected in the title as well.

Methodology:

• In Table 1, it may be more effective to present the 13 schools separately rather than together.

• Table 1 specify years for age.

• Instead of focusing on theoretical scenarios, it could be more realistic to ask schools about the eye care provided to the children currently under their care. A theoretical methodology is likely to lead to positive subjective bias.

• The number of school managers who are aware of children with an eye condition does not mean that they have identified all the children with eye conditions if a full sight test not carried out on all children.

• Stating that health programmes are warranted twice a year does not mean they occur, particularly for all children. Many people are under the impression that some children are too disabled for a sight test. What is included in a health programme?

• It would be valuable to determine the percentage of children who have received an eye test or screening program. Which type was it, how long ago it occurred, and the average age at the first test.

• The study should also inquire if each child has a documented sight test in their clinical records and if there is a management plan for any diagnosed conditions.

• Comparing the number of visually impaired children, those wearing refractive correction, or diagnosed with eye conditions to other populations of children with special educational needs (SEN) would provide useful insights. If significantly lower numbers are found, it suggests that schools are not identifying all cases.

• It would be beneficial to inquire if carers understand the implications for each child diagnosed with an eye condition and if they felt they required further education. For example, clarification could be sought regarding whether a child with freely alternating manifest strabismus would be classified as visually impaired and require magnification or if it simply affects their coordination.

• It would be pertinent to ask how often each child's glasses should be checked and what the procedure is in case of breakage.

• Online questionnaire should be included.

Results:

• It is unclear why there is a discrepancy in the responses of participants 1 and 2 regarding the frequency of eye check-ups (yearly or on admission). Clarification should be sought from all schools to confirm if children receive an annual eye health assessment or one on admission, as stated by the health organizers. Additionally, it would be important to determine if any children fall through the gaps and do not receive an eye test.

• It would be relevant to explore if schools are aware that they can request an SEH program, as stated by the organizers.

• The schools' perspectives on the barriers stated by the organizers should be included. Do they feel unwilling to participate?

• It should be clarified if the schools are allowed to refer children to a doctor and, if so, which type of doctor.

• Environmental modifications or adaptations mentioned in the results should be explained.

• The SEH organizers could be asked if the identified barriers impact the child's annual or admission eye check.

• If the organizers believe that SEH is unnecessary because eye care already occurs, it raises the question of why they are concerned about barriers to SEH.

Discussion:

• When children have difficulties with reading, writing, and multiple disabilities, it can be challenging for caregivers to determine whether the issues stem from eye problems, IQ problems, autism, or other factors related to their disabilities. It would be helpful to address whether all children struggling are referred to eye specialist even if carers are unsure what the problem is.

• The SEH organizers should be checked whether they have verified if each child receives their annual or admission eye check-up, or if they assume that it happens.

Reviewer #3: This is an interesting and useful contribution to understanding the barriers to service provision in special schools.

My comments:

1) Line 81-84. In the introduction the authors refer to eye care programs in Wales. There is now a wider England study that is driving the debate that I think is worth considering.

Findings from an opt-in eye examination service in English special schools. Is vision screening effective for this population? L. A. Donaldson, M. Karas, D. O’Brien and J. M. Woodhouse. PloS one 2019 Vol. 14 Issue 3 Pages e0212733

I would also like to see this idea explored more as to the relevance or not of this single example of a program in the UK.

2) The study defines itself as a mixed methods study (line 98-99) but it is hard to work out how the data was handled? It talks of a thematic analysis and I am assuming this is of both the open text parts of the online questionnaires and the recorded interviews. (Line 148-148). This needs to be clarified. It then goes onto to say the questionaire data themes were determined but says nothing of the interview thematic analysis which you would expect to lead to some other themes.

In the results section it then discusses results under each of these three themes defined in the methods section but this data seems to be only the results of knowledge, practice and attitude multiple choice questions. I cannot see any discussion of the results of the thematic analysis of the free text responses.

3) The results of the interviews do offer quotes as would be expected of a thematic analysis and the data does seem to be grouped into headings but there is no mention made of themes generated form an analysis of this data.

4) There is no discussion of the methodology of the thematic analysis. How was it planned, how was the data coded and how was a thematic framework arrived at. There is also no mention of the theoretical underpinnings of the approach to the thematic analysis. This is a complex subject but I feel if the authors a claiming a thematic methodology they should make some effort to describe within the bounds of accepted thinking on this method. (Braun & Clarke, 2006; Kiger & Varpio, 2020). This does not need to be detailed but it does need a mention.

Braun, V., & Clarke, V. (2006). Using thematic analysis in psychology. Qualitative Research in Psychology, 3(2), 77-101. doi:10.1191/1478088706qp063oa

Kiger, M. E., & Varpio, L. (2020). Thematic analysis of qualitative data: AMEE Guide No. 131. Med Teach, 42(8), 846-854. doi:10.1080/0142159X.2020.1755030

6. PLOS authors have the option to publish the peer review history of their article (what does this mean?). If published, this will include your full peer review and any attached files.

**Do you want your identity to be public for this peer review?** For information about this choice, including consent withdrawal, please see our Privacy Policy.

Reviewer #1: No

Reviewer #2: No

Reviewer #3: No

---

## [Decision Letter · Decision Letter 1]

27 Feb 2024

PGPH-D-23-01037R1

A cross-sectional survey of eye health knowledge, attitude, and practice among special school managers and barriers to eye health programmes in special schools in Hyderabad, India

Dear Dr. Devaraj,

Thank you for submitting your manuscript to PLOS Global Public Health. After careful consideration, we feel that it has merit but does not fully meet PLOS Global Public Health’s publication criteria as it currently stands. Therefore, we invite you to submit a revised version of the manuscript that addresses the points raised during the review process.

We thank you for having taken the time to engage with the comments from the first round of reviewers and acknowledge that the corrections were made in an attempt to try to address required clarification with relevant inclusions and deletions as suggested.

However, as noted in the responses from the latest round of reviewers, the challenges still remaining specifically relate to weaknesses in the study methods. The 2 SEH participants represent organizations which engage with a small minority of special schools. This raises questions as to whether the cited barriers also apply to the organizations not included, which represent the majority of schools in the study area. The absence of a rigorous data analysis process for the qualitative part (barriers) of the study is a major flaw. Having insight into the identified codes, with any sub categories and the iterative/inductive steps that were applied to data that eventually lead to the emergent themes will provide clarification on the themes which emerged directly from the collated data. Presenting the overlapping in codes and how these were managed is unclear and hence, reviewers were unable to determine exactly how the 7 themes reported under barriers emerged. The results presented in this section cannot be generalized due to the small sample size and very limited school representation by the respective organizations.

The open-ended sections from the survey should be coded and grouped into themes and sub-themes rather than being presented as an additional section in Table 2. Previously cited themes are missing or different to that reported on in the last version of the paper. There is an additional concern, as raised by a reviewer, that there are now some results that are introduced in the Discussion which are not reflected in the Results. This warrants very careful review of the corrections applied and ensuring that there is no ew data included.

The discussion does not address the the fact that SSM's cited academic difficulties as major challenges experienced by children with special needs, yet advise that the children should be integrated into mainstream schools. It will be expected that the integration may cause more significant academic demands. This is not adequately addressed. There should be no results presented in the discussion section and statements emanating from participants must be separated from opinions of researchers.

These issues will need to be addressed so that the paper may be considered for acceptance. 

However, please review the additional comments section to find my overall recommendation on this paper. 

We look forward to receiving your revised manuscript.

Kind regards,

Vanessa Raquel Moodley, PhD

Guest Editor

Journal Requirements:

1. Please provide additional details regarding participant consent. In the ethics statement in the Methods and online submission information, please ensure that you have specified (1) whether consent was informed and (2) what type you obtained (for instance, written or verbal, and if verbal, how it was documented and witnessed). If your study included minors, state whether you obtained consent from parents or guardians. If the need for consent was waived by the ethics committee, please include this information.

Additional Editor Comments (if provided):

Noting that the major weakness of the paper lies in the Qualitative interview component of this paper, I suggest that researchers consider only reporting on the SSM questionnaires in this paper, incorporating the recommendations of the reviewers.

The sample for the interviews could be increased to secure better representation of the organizations and validity of "best practices" recommendations. Further, thematic data analysis will need to be conducted in a significantly more rigorous manner after which that article could be written and submitted to a suitable journal for consideration for publication.

Reviewers' comments:

Reviewer's Responses to Questions

Reviewer #4: All comments have been addressed

Reviewer #5: (No Response)

Reviewer #6: All comments have been addressed

2. Does this manuscript meet PLOS Global Public Health’s publication criteria? Is the manuscript technically sound, and do the data support the conclusions? The manuscript must describe methodologically and ethically rigorous research with conclusions that are appropriately drawn based on the data presented.

Reviewer #4: No

Reviewer #5: Yes

Reviewer #6: Partly

3. Has the statistical analysis been performed appropriately and rigorously?

Reviewer #4: (No Response)

Reviewer #5: Yes

Reviewer #6: Yes

4. Have the authors made all data underlying the findings in their manuscript fully available (please refer to the Data Availability Statement at the start of the manuscript PDF file)?

Reviewer #4: Yes

Reviewer #5: Yes

Reviewer #6: Yes

5. Is the manuscript presented in an intelligible fashion and written in standard English?

Reviewer #4: Yes

Reviewer #5: Yes

Reviewer #6: Yes

6. Review Comments to the Author

Reviewer #4: • The sample size of 33% was found to be too low, indicating a low uptake.

• The small size is too low for the results to be generalized

• 2/4 invited eye health programme organizers, although they form 50% of the population an effort could have been made to interview online all 4 health programme organizers

COMMENT:

Even though the findings of the study were found to be valuable, the study seems like it is a pilot study project.

Reviewer #5: MY COMMENTS

GENERAL COMMENTS

This is an interesting study and worth reporting. The manuscript is very well written with least grammatical errors. However, some of the underlisted limitations are ones authors could have corrected easily before commencing the actual data collection. Example is especially ensuring the questionnaire used was reliable and valid to measure exactly what it seeks to measure. I think authors instead of recommending another study to report on a comprehensive situational analysis, should have taken some time to collect data and analyze these findings also to enrich the current report further.

SPECIFIC COMMENTS

Line 39-40 on the old document ; Lines 41 -42 of revised document: Is not grammatically correct to begin a sentence with a number or fraction like this. I suggest authors modify this sentence. E.g. A number of thirteen (13) out of sixty-seven (67) invited SSM (19%) participated.

Line 51 on the old document; Lines 53-54 of revised doc: I think current work should have considered also conducting a comprehensive situational analysis of these findings so discussion would be rich. This is just a scanty result.

Lines 128-130 in old document; Lines 146 – 148 on the revised doc: How are authors sure responders’ answers reflect their current understanding. What precautions or measures did author put in place to ensure responders to not refer to other resources on the internet especially as questionnaire stayed with them for two weeks. Please indicate this.

Line 154-155 in old document; Line 172 -173 on revised document: Demographic data like age and gender distributions are repeated in tables and in the text. It will be advisable for authors to delete from the text and to simply refer to the table 1 to avoid repetitions.

Line292 in old document; Line 445 -447 I revised document: The questionnaire used should have at least been tested for its reliability. If these was done in the pilot study, please indicate how reliable this tool was. With use of an unreliable questionnaire, how are we sure of the reliability of the results?

Reviewer #6: General comments:

I appreciate the author's efforts to explore barriers to eye health service provision in special schools in Hyderabad India. The study investigates the “Knowledge, attitude and practice” of special school headteachers regarding various aspects of eye health among children in their schools. Further, the study also reports from interviews conducted among managers of two organisations involved in school eye health programs in the area.

The findings from the study could be of interest to those working in this field, particularly for those involved in program planning and management in the local area. The authors have attempted to address the comments raised by the reviewers in the first round.

The manuscript in its current form, however, still has several concerns and limitations. As previous reviewers raised, there is significant concern about the validity and comprehensiveness of the survey questionnaire used. Further, the limited sample size (both quantitative and qualitative parts) also hinders any generalisation of the findings. The language and formatting of the manuscript also need improvement for clarity, conciseness, and consistency.

I suggest that the authors undertake a thorough revision and subsequently resubmit the work for further consideration.

Specific comments:

1. Title:

a. Considering the length of the title, the study design could be removed (as the study not only included “a cross-sectional survey” but also interviews).

b. Also consider clarifying that the barriers relate to organising the eye health programs (not accessing). For example, “…. and barriers to organising eye health programs…”

Abstract:

1. Background: the aims could be revised for clarity and ease of reading.

2. Methods:

a. Were the questions focused on “visual needs” or “eye health” as in the title?

3. Results:

a. Line 43-45: needs to be revised for clarity

b. Line 45-46: “Only 23% of special schools organized yearly SEH programmes.” But an additional 8% (1) school organised it twice a year. So, 31% organized at least once a year?

c. I suggest the authors be consistent with how the barriers are reported here and in the main results section.

4. Conclusion:

a. Consistency in the use of “visual needs”, “eye health” or “eye health needs” as the focus of the study.

b. I suggest the authors be consistent with how the barriers are reported here and in the main results/discussion/conclusion section.

Introduction:

1. Line 70-71: “..SEN are at a higher risk of developing VI…”. This suggests that being in a SEN is causing VI (as diabetics have a higher risk of vision impairment). Or is it that these children have a higher prevalence of vision impairment and ocular comorbidity?

2. Add a description of what school eye health programs are in general and in the local context (India/Hyderabad).

3. Line 98: Should add the affiliation of this person

4. Line 99-108: The rationale for focusing on KAP of school managers (rather than the teachers or carers who are directly involved in the daily care of the children) needs to be clearly and explicitly described.

Methods:

1. Line 126-129: How were these organisations identified? Are the investigators confident that all organisations were identified?

2. Line 137: What is the rationale for including an experience limit for the inclusion of the SHE program organizers?

3. Line 163-166: the revised data analysis segment provides additional information on the ‘thematic analysis’ citing Braun and Clark 2006 in response to the previous reviewer’s comments. I am just wondering if the authors used any software to manage the data, followed all the steps as outlined by Braun and Clarke (2006), were the codes inductively generated or any theoretical frameworks used, and if there was an iterative refining and defining of the themes. Reporting on the investigator's roles and backgrounds in this process and providing the above details will help strengthen the qualitative component (adding reflexivity and credibility) of the study.

Results:

1. Mention that the sample in this study represents 13 of 67 schools (19%)

2. Information in Table 1 is largely provided in the text above, so it could be removed to reduce the length of the paper.

3. For open-ended questions, clearly mention that the responses are the author’s codes/subthemes and not the verbatim responses of the participants. It also needs to be clarified that some participants’ responses may have been coded to more than one code/subtheme.

4. The results presented in Lines 184-190 do not correspond exactly to those in Table 2 as they are not reported consistently (text shows a theme of Difficulties faced in performing academic tasks” including two rows of the table. The same goes for the rest of the text/table rows. The themes and codes/subthemes within each theme need to be reported explicitly and consistently in both text and table.

a. It would be important if any participants were not aware of the symptoms of vision problems, but this information is hard to make from the data presented.

5. The methods section mentions categorising themes as good/poor knowledge, positive/negative attitude and acceptable/non-acceptable practices. This is not reported in the results section.

a. I also if all the responses or subthemes were categorised into at least one category. For example, in Table 4, how were the responses to the question “What would you do If you suspect a child to have eye problem? (Open ended)” categorised into acceptable or non-acceptable practices?

b.

6. Rather than lengthy paragraphs discussing multiple concepts/questions, please use paragraph breaks.

7. Table 2: “How often is eye check-up warranted for children SENs”. What would be the rationale for putting this question as an MCQ and not an open-ended question? And what is the rationale for the limited options presented (for example, not including a 6-monthly option)? The same goes for the rationale for selecting specific MCQs for other questions as well.

8. Line 207: “To the question asked in open text boxes about why/why not children with VI should be attending mainstream schools, …” Was that the question asked?

9. Normal school vs regular school: do they mean the same thing?

10. Table 4:

a. Check the total number in the right column. I can understand there may be multiple codes/subthemes emerging from the response of one participant (and this needs to be explicitly mentioned) but this still does not explain why some questions only have 12 responses (including “did not answer” and “others”).

b. For the rows with “others” and just 1 (8%), please spell out what that response was.

11. Interview responses

a. I was surprised that there were only “four known community outreach organizations and more than that, I was disappointed that participation from all of them was not secured.

b. Based on the SSM survey responses, about a quarter of the schools reported organising eye health programs at least once a year. If we assume this to be a representative sample from the 67 special schools in the area, about 17 schools would conduct annual eye health programs. However, the organisations included in the interview only covered 5+1 special schools (Line 251-252). Does that mean the remaining two organisations that could not be interviewed covered most of the special schools? That would be a significant limitation of the study as the key organisations providing most of the eye health programs to special schools were missed.

c. I’d encourage authors to explicitly mention or clarify at the beginning of this section (and wherever relevant) that the statements presented within the themes are as expressed by the participants. Sometimes it may be confused with the author's opinion. For example, are the following statements based on what the data (participants' responses) suggests or a fact or opinion of the authors?

i. “Special SEH programmes are usually done as stand-alone programmes and hence they are rarely included in government or NGO sponsored activities.”

ii. “ … special SEH programmes require the parents to attend while the eye examination is being carried out.”

iii. “Most SEH programmes are conducted by community eye health workers, schoolteachers and vision technicians who are not trained to carry out a complete eye examination.”

d. Reading the description of the first barrier (low program coverage) for organisers, it feels like this theme is more about low priority (which then resulted in low coverage). This is my impression reading the description, not the full data. That’s why I reiterate my earlier comments about iterative refining and defining/naming of the themes and the importance of reporting on the investigator's roles in this process.

e. I also note that the themes in the revised manuscript are different to the barriers listed in Box 2 of the original submission. Some of the previously mentioned barriers such as “no standard screening protocols” and “infrastructures are inaccessible” are not mentioned as barriers in the revised submission.

f. Line 267-268: “On the other hand, special schools seldom take the initiative in organizing eye health programmes.” If this is the organisers’ perception, were they probed on why they believed so?

g. A lack of stakeholder engagement: were the organisers probed about the engagement or motivation of the school management? Considering the earlier part of the study on SSM and their potential role in organising eye health programs, it seems important to explore this aspect.

Discussion:

1. The discussion section as such is quite lengthy and needs to be a bit more concise.

2. Some of the statements in the discussion seem to better fit the results section or are repeated from the results section.

3. Some results are introduced in the discussion without a mention in the results section. For example:

a. Line 381-382: “They also felt studying with normal peers could improve their psychosocial well-being.”

b. Line 400: “However, this was not reported by the SSM staff.” SSM’s responses relating to annual health check-ups covering eye examinations were not mentioned in the results.

4. Line 354-357: Does this mean that the children haven’t had any eye examinations in past? It is important to have an eye examination, but it does not need to be through a school eye health program.

5. The discussion around Vitamin A and its deficiency seems overstretched as this was how the survey questions were designed (MCQ without Vit A option!). Further, Gogate at el. (Gogate P, Soneji FR, Kharat J, Dulera H, Deshpande M, Gilbert C. Ocular disorders in children with learning disabilities in special education schools of Pune, India. Indian J Ophthalmol. 2011 May-Jun;59(3):223-8.) shows that about 2% of bilateral vision impairment is due to vitamin A deficiency in children with learning disabilities in special education schools of Pune, India.

Limitations:

1. While the sample size limitation is mentioned, authors need to clearly mention its implication i.e. the generalisability of findings. Further, the small number of interviews conducted needs to be mentioned with its implications in data saturation and confidence in the completeness of the barriers unearthed.

Language and formatting:

Some suggestions to improve the language and formatting of the manuscript:

1. Break down the long and complex sentences with multiple concepts. For example, the first sentence in the abstract.

2. Minimise the use of non-standard abbreviations

3. Consistent use of decimal points (30.8% vs 31%) and terms (eye health providers vs organisers)

4. Careful re-examination of the numbers and proportion reported. E.g.,

a. Line 359: “…only less than a third of participants were aware of this condition.” Table 2 shows 23% which is less than a quarter of the participants.

b. Line 362: “All participants said poor nutrition…” Table 2 shows 12 of 13 participants for this response.

7. PLOS authors have the option to publish the peer review history of their article (what does this mean?). If published, this will include your full peer review and any attached files.

**Do you want your identity to be public for this peer review?** For information about this choice, including consent withdrawal, please see our Privacy Policy.

Reviewer #4: **Yes: **Thokozile Ingrid Metsing

Reviewer #5: No

Reviewer #6: No

---

## [Editor Report · Decision Letter 2]

27 May 2024

PGPH-D-23-01037R2

Eye health knowledge, attitude, and practice among special school managers and barriers to eye health programmes in special schools in Hyderabad, India

Dear Dr. Prakash,

Thank you for re-submitting the manuscript to PLOS Global Public Health. Well done on carefully addressing each of the comments and incorporating the suggestions from previous reviews. After careful consideration, we feel that, as it currently stands, there are still minor revisions needed for it to fully meet PLOS Global Public Health’s publication criteria . Therefore, we invite you to submit a revised version of the manuscript that addresses the minor points raised during the review process.

The remaining minor edits required are tracked in the attached manuscript.

It is further recommended that on completion of the revisions, you seek assistance of a language editor to attend to the language edits needed.

We look forward to receiving your revised manuscript.

Kind regards,

Vanessa Raquel Moodley, PhD

Guest Editor

Journal Requirements:

1. Please provide additional details regarding participant consent. In the ethics statement in the Methods and online submission information, please ensure that you have specified (1) whether consent was informed and (2) what type you obtained (for instance, written or verbal, and if verbal, how it was documented and witnessed). If your study included minors, state whether you obtained consent from parents or guardians. If the need for consent was waived by the ethics committee, please include this information.

Additional Editor Comments (if provided):

You have successfully achieved the objective of identifying the gaps in eye health services to children with SEN and raised awareness towards better care in the sector.
---

## [Editor Report · Decision Letter 3]

9 Jul 2024

PGPH-D-23-01037R3

Eye health knowledge, attitude, and practice among special school managers and barriers to eye health programmes in special schools in Hyderabad, India

Dear Dr. Devaraj,

Well done on attending to the previously suggested revisions. There are very minor edits required as follows:

1. When presenting the qualitative results you must clearly indicate that the summarized statements included are those of interviewees. As it stands, in a few instances, it sometimes appears to be sentiments of the authors.

2. Conduct a final read through to ensure all scholarly language edits are attended to.

The amended article should be returned by the 21st July 2024.

Reviewers' comments:

Journal Requirements:

1. Please provide separate figure files in .tif or .eps format only and remove any figures embedded in your manuscript file. Please also ensure all files are under our size limit of 10MB.

---

## [Editor Report · Decision Letter 4]

22 Jul 2024

Eye health knowledge, attitude, and practice among special school managers and barriers to eye health programmes in special schools in Hyderabad, India

PGPH-D-23-01037R4

Dear Mr Devaraj,

We are pleased to inform you that your manuscript 'Eye health knowledge, attitude, and practice among special school managers and barriers to eye health programmes in special schools in Hyderabad, India' has been provisionally accepted for publication in PLOS Global Public Health.

Best regards,

Vanessa Raquel Moodley, PhD

Guest Editor
